# WAVEDIFFUSION: JOINT LATENT DIFFUSION FOR PHYSICALLY CONSISTENT SEISMIC AND VELOCITY GENERATION

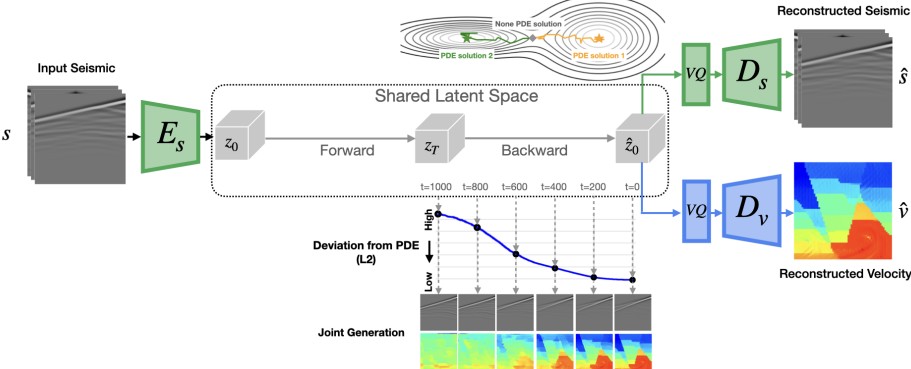

Figure 1: **Overview of WAVEDIFFUSION**. WAVEDIFFUSION generates paired samples that satisfy the governing PDE via a joint latent diffusion framework. The diffusion guides the sampling process toward physically valid solutions through denoising, progressively mapping non-solution points (gray squares in valleys) to valid solutions (colored stars at peaks).

## ABSTRACT

Full Waveform Inversion (FWI) is a critical technique in subsurface imaging, aiming to reconstruct high-resolution subsurface properties from surface measurements. Acoustic FWI involves two physical modalities, seismic waveforms and velocity maps, which are governed by the acoustic wave equation. Prior works primarily focus on the inverse problem, modeling the relationship between seismic and velocity as an image-to-image translation task. In this work, we study their relationship from a generative perspective. Our aim is to explore and characterize the latent space structure, and identify latent vectors that generate seismic–velocity pairs consistent with the governing partial differential equation (PDE). Specifically, we model seismic and velocity data jointly from a shared latent space via a diffusion process. In experiments, we find that diffusion progressively refines arbitrary latent vectors into ones that yield approximately physics-consistent seismic–velocity pairs, even without explicit physics constraints. This provides empirical evidence of PDE-consistency in latent diffusion, where sampling is biased toward PDE-valid solutions. In latent space, satisfying the acoustic wave equation can be approximated through sampling and gradient descent. We formalize this physics-consistent latent modeling task and quantify it through extensive experiments. On large-scale OpenFWI benchmarks, our approach produces high-fidelity, diverse, and physically consistent seismic–velocity pairs, demonstrating the potential of a data-driven latent diffusion for physically consistent generation in a complex scientific domain.

## 1 INTRODUCTION

Subsurface imaging is critical in scientific and industrial applications, including earthquake monitoring Virieux et al. (2017); Tromp (2020), greenhouse gas storage Li et al. (2021); Wang et al. (2023b), medical imaging Guasch et al. (2020); Lozenski et al. (2024), and oil and gas exploration Virieux & Operto (2009); Wang & Alkhalifah (2018). At its core, Full Waveform Inversion (FWI) involves

reconstructing velocity maps that describe the propagation speed of seismic waves, governed by the acoustic wave equation:

$$\frac{\partial^2 s(\mathbf{x}, t)}{\partial t^2} = v^2(\mathbf{x})\nabla^2 s(\mathbf{x}, t) + S(\mathbf{x_s}, t), \tag{1}$$

where $s(\mathbf{x}, t)$ is the seismic data, $v(\mathbf{x})$ is the velocity model, $\nabla^2$ is the Laplacian, and $S(\mathbf{x_s}, t)$ is the source term. Most recent machine learning (ML) approaches have focused on the FWI problem, which formulates this task as an inverse problem. For example, supervised-based methods frame the relationship between seismic and velocity as an image-to-image translation task Wu & Lin (2019); Zhang et al. (2019); Sun & Demanet (2020); Feng et al. (2021).

In this work, we revisit the relationship between seismic and velocity from a generative perspective. While seismic and velocity data are connected by the wave equation, their distribution in the original seismic-velocity joint space is unknown. Rather than treating their relationship purely as an inversion task, our goal is to explore and characterize the latent space structure shared by both modalities. Our preliminary study reveals that randomly sampled latent vectors often decode into pairs that violate the PDE 1, suggesting only a sparse subset of the latent space corresponds to physically valid solutions. Naturally, this raises the question: *Can we systematically identify physically valid latent space representations that adhere to the governing PDE from those that do not?*

Our key finding is that diffusion models naturally enforce physical consistency when applied in a shared latent space. By jointly modeling seismic data and velocity map from a shared latent space via a diffusion model, the denoising process progressively transforms arbitrary latent sampling points into ones that decode into solutions more consistent with the governing PDE, as illustrated in Figure 1. This indicates that, even without the need for explicit physical constraints, a purely data-driven latent diffusion *implicitly* biases sampling toward physically valid regions. In other words, in the latent space, approximate satisfaction of the acoustic wave equation can emerge through sampling and gradient descent. Such behavior cannot be assumed a priori and highlights an emergent property of latent diffusion. This data-driven approach is particularly attractive when the forward model is unavailable, non-differentiable, or computationally prohibitive, offering a practical alternative.

Differing from recent diffusion-based approaches to inverse problems (Song et al., 2022; Chung et al., 2023; Zhang et al., 2025), which recover one modality from the other (e.g., velocity from seismic), our objective focuses on the physics-consistent latent modeling task. The goal is to identify latent vectors that jointly generate both modalities in a manner consistent with the governing PDE. More broadly, our work is an exploration into the intersection of generative modeling and scientific data modeling, demonstrating both the potential and the limitations of current approaches.

On large-scale OpenFWI Deng et al. (2022) benchmarks, we evaluate our approach through extensive experiments, demonstrating it produces high-fidelity, diverse, and physically consistent seismic–velocity pairs. Furthermore, we show that these jointly generated pairs can augment training data for conventional supervised models such as InversionNet Wu & Lin (2019), thereby validating both their quality and practical utility.

## 2 WAVEDIFFUSION: EXPLORING LATENT SPACE VIA JOINT GENERATION

In this section, we introduce WAVEDIFFUSION, a framework for jointly modeling seismic and velocity data in a shared latent space. Specifically, we first formalize this physics-consistent latent modeling task. Then, we investigate two fundamental questions: (1) Do all latent representations produced by conventional image-to-image models satisfy the governing PDE? (2) If not, can we distinguish latent samples $\{z\}$ that yield physically consistent outputs from those that do not? To address these questions, we adopt a two-stage latent diffusion framework that aligns naturally with our objective. First, we construct a latent space using an encoder-decoder model. Then, we apply a latent diffusion process to refine latent codes toward physically valid solutions progressively.

### 2.1 PHYSICS-CONSISTENT LATENT MODELING

We now formalize the notion of the physics-consistent latent modeling task.

Let $f : \mathcal{V} \to \mathcal{S}$ denote the forward modeling operator (e.g., a numerical wave simulator), mapping a velocity field $v \in \mathcal{V} \subset \mathbb{R}^{H \times W}$ to its corresponding seismic response $s \in \mathcal{S} \subset \mathbb{R}^{T \times W}$, i.e., $s = f(v)$.

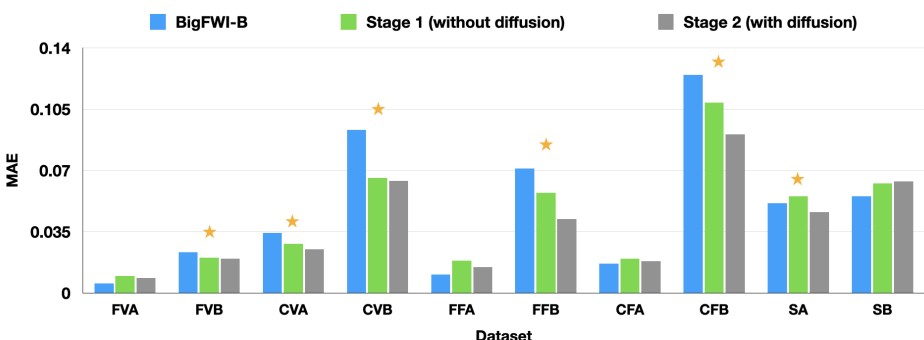

Figure 2: **Comparison of inversion performance.** Mean Absolute Error (MAE) across various OpenFWI datasets for encoder-decoder-based models and diffusion refinements. Our models perform competitively against BigFWI-B, with diffusion providing slight refinements. Yellow stars indicate datasets where our model outperforms the BigFWI-B baseline.

Let $z \in \mathcal{Z} \subset \mathbb{R}^d$ denote a latent representation in a learned space. We assume the existence of two deterministic decoders: $D_v : \mathcal{Z} \rightarrow \mathcal{V}, \quad D_s : \mathcal{Z} \rightarrow \mathcal{S}$, where $D_v$ maps latent vectors to velocity fields and $D_s$ maps latent vectors to seismic signals.

A latent vector $z \in \mathcal{Z}$ is defined to be *physically valid* if it satisfies the following condition:

$$\|D_s(z) - f(D_v(z))\| < \epsilon,$$

for some tolerance $\epsilon > 0$. In words, the seismic signal decoded from $z$ must be close to the one obtained by applying the forward operator to the decoded velocity. The central challenge is thus to characterize or sample from the subset of the latent space that satisfies this condition:

$$\mathcal{Z}_{\text{valid}} := \{z \in \mathcal{Z} \mid \|D_s(z) - f(D_v(z))\| < \epsilon\}.$$

Two fundamental properties of this problem are of particular interest:

- **Completeness.** To what extent can the valid set $\mathcal{Z}_{\text{valid}}$ be captured? That is, what fraction of physically valid solutions are representable under the learned distribution $p(z)$?
- **PDE Satisfaction.** How small can the residual error $\epsilon$ be made in practice, and what factors (e.g., model capacity, dataset coverage, optimization dynamics) govern this error bound?

In practice, as we will discuss in the following, latent vectors sampled arbitrarily from $\mathcal{Z}$ typically fail to meet this criterion, which motivates the need to learn a structured distribution $p(z)$ over $\mathcal{Z}_{\text{valid}}$.

## 2.2 STAGE 1: ENCODER-DECODER AND RECONSTRUCTION

**Extending encoder-decoder by adding reconstruction branch:** To enhance the interpretability of the latent space, we extend conventional encoder-decoder FWI models by incorporating an additional seismic decoder branch. This modification enables the simultaneous reconstruction of both seismic and velocity maps from a shared latent representation, facilitating a deeper analysis of its structure. This enables the analysis of whether all latent space points satisfy the PDE. To facilitate the second-stage diffusion process, we incorporate vector quantizations in the latent space.

**Achieving comparable performance in FWI:** We evaluate the inversion performance of our vector-quantized encoder-decoder model against BigFWI-B Jin et al. (2024), a state-of-the-art InversionNet trained on the full OPENFWI dataset. This provides a fair comparison, as the training sample volume and model parameter size of BigFWI-B (24M) approximately align with our Stage 1 model (19M). Figure 2 shows that our encoder-decoder model achieves competitive performance relative to BigFWI-B across all datasets, outperforming BigFWI-B in six out of ten datasets (marked with yellow stars).

**Definition of deviation from PDE:** To quantitatively investigate whether randomly sampled $z$ points satisfy the PDE, we measure the deviation of the generated seismic-velocity pairs from the governing PDE. Specifically, for a randomly sampled latent representation $z$, we decode the seismic data $\hat{s}$ and velocity map $\hat{v}$. Using a finite difference solver, we compute the ground truth seismic $s_{\hat{v}}$ for the generated $\hat{v}$. The deviation from the PDE is quantified as the L2 distance $\|\hat{s} - s_{\hat{v}}\|_2$.

**Most latent points do not correspond to PDE solutions:** The deviation of decoded modalities from randomly sampled $z$ points, which are usually far away from trained latent points, reaches an average L2 distance of 0.013 (Figure 5 backward step 0), which is eight times higher than the deviation (0.0016, Figure 5 forward step 0) of reconstructed modalities from the training set (trained latent points). Figure 3 visually compares the generated samples from Stage 1 (row 1) and the original OpenFWI dataset (row 3). The encoder-decoder provides coarse approximations of PDE solutions but does not fully satisfy the wave equation.

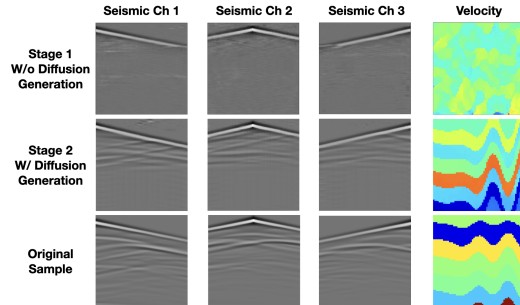

To address whether poor reconstructions result merely from latent distribution mismatch, we conduct additional experiments, detailed in Appendix A.3, and further confirm that the diffusion model significantly improves reconstruction quality. Encoding and decoding through this latent space alone do not inherently ensure physical validity. Thus, most generated samples fail to satisfy the governing PDE, meaning only a sparse subset of $z$ corresponds to valid solutions.

Figure 3: Generated samples Comparison.

### 2.3 STAGE 2: JOINT DIFFUSION IN THE LATENT SPACE

**Refining the latent space with diffusion:** Since random sampling in the latent space does not guarantee adherence to the PDE, we use a *joint diffusion model* to transform arbitrary latent points into valid ones progressively. This process follows a standard forward-backward diffusion formulation:

**Forward process:** Gaussian noise is added to the latent vector $z$, creating a noisy representation: $z_t = z_{t-1} + \epsilon_t, \quad t = 1, \ldots, T$, where $\epsilon_t$ is the noise applied at step $t$, with in total $T$ steps.

**Backward process:** The noisy latent vector is progressively denoised through the reverse diffusion process: $z_{t-1} = z_t - \gamma_t \nabla_{z_t} \mathcal{L}(z_t, t)$, where $\gamma_t$ is a step size, and $\mathcal{L}$ is the denoising objective:

$$\mathcal{L} = \mathbb{E}_{z_0, \epsilon \sim \mathcal{N}(0, I), t} \left[ \|\epsilon - \epsilon_\theta(z_t, t)\|_2^2 \right]. \tag{2}$$

This follows the standard formulation of denoising score matching for latent diffusion models Rombach et al. (2022). Here, $\epsilon_\theta(z_t, t)$ is the predicted noise estimation. The loss function minimizes the discrepancy between the true noise $\epsilon$ and the model's predicted noise $\epsilon_\theta(z_t, t)$, guiding the model to iteratively refine the noisy latent variable back to a valid solution in the latent space.

Once trained, the model can generate new seismic-velocity pairs that satisfy the wave equation by sampling latent vectors from a Gaussian distribution and refining them via the learned backward process: (1) Sample a latent vector $z_t$ from a standard Gaussian distribution: $z_t \sim \mathcal{N}(0, I)$. (2) Pass $z_t$ through the backward denoising steps: $z_{t-1} = \mathcal{L}(z_t), \quad t = T, \ldots, 1$. (3) Decode $z_0$ back into seismic data and velocity maps: $\hat{s} = D_s(z_0), \ \hat{v} = D_v(z_0)$.

### 2.4 INSPECTING DIFFUSION PROCESS

To analyze the role of diffusion, we measure the deviation of generated seismic-velocity pairs from the governing PDE at each diffusion step using the L2 distance between $\hat{s}$ and $s_{\hat{v}}$. Figure 4 visualizes this process, while Figure 5 presents the statistical evaluation on 13.2k samples at each diffusion step.

**Deviation increases by adding noise:** In the left half of Figure 4, during the forward diffusion process, as noise is added, the seismic data $\hat{s}$ generated by the model diverges more from the ground truth $s_{\hat{v}}$. This divergence, shown as the channel-stacked difference in the last row, reflects the increasing deviation from PDE as the noise level rises. Meanwhile, the decoded modalities show distortion in structures. The statistical evaluation in Figure 5 illustrates how the deviation increases, from the initial deviation 0.0016 (trained latent points) to 0.0121 (pure noise points).

**Deviation decreases by denoising:** In contrast, the right half of Figure 4 shows the backward process, where noise is progressively removed, reducing the deviation and refining the seismic-velocity pairs toward solutions with smaller stacked seismic difference and restored structures. Similarly, the statistical evaluation in Figure 5, the deviation decreases as the generated pairs are refined into physically valid solutions from an averaged L2 distance of 0.013 back to 0.002, confirming the model's ability to refine latent points toward physically consistent solutions.

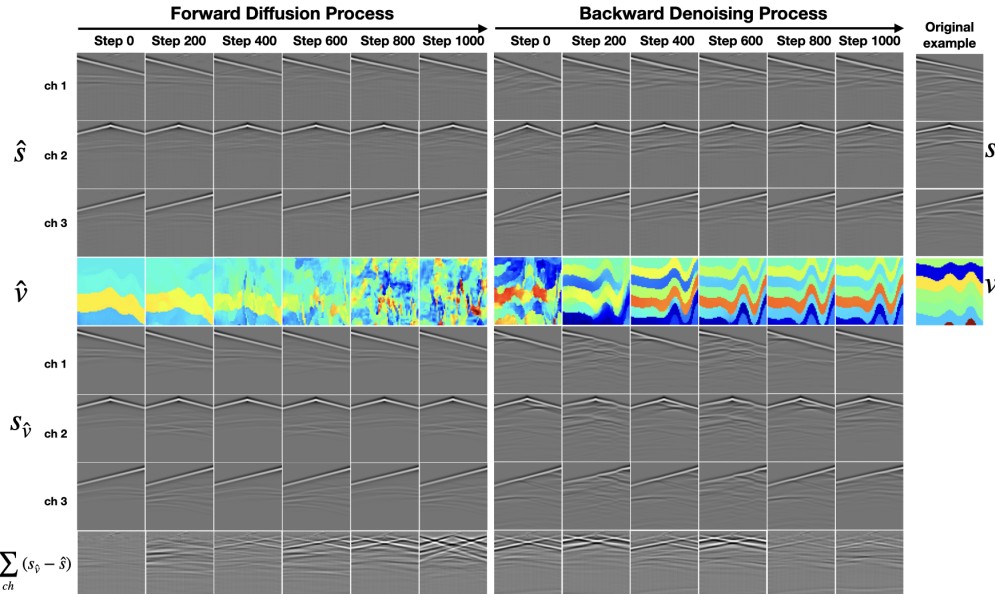

Figure 4: **Visualization of deviation from PDE during diffusion.** Seismic data of a CVB example at different timesteps during the forward (left half) and backward diffusion processes (right half). Rows 1-3 show generated seismic channels $\hat{s}$ by the joint diffusion model. Row 4 shows the generated velocity map $\hat{v}$. Rows 5-7 show the ground truth seismic data $s_{\hat{v}}$ calculated for the generated $\hat{v}$. Row 8 shows the deviation, visualized as the channel-stacked difference between $s_{\hat{v}}$ and $\hat{s}$. Noise increases the deviation during the forward diffusion, and the reverse process reduces discrepancies.

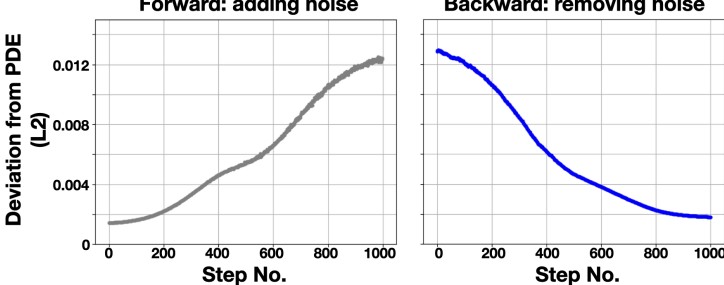

Figure 5: **Deviation from the governing PDE.** The L2 distance is calculated between generated seismic data $\hat{s}$ and ground truth $s_{\hat{v}}$ of the generated velocity map $\hat{v}$ using a finite difference solver.

Our results reveal that only a small subset of latent points satisfies the PDE, while diffusion progressively moves high-deviation points toward physically valid solutions. This suggests that diffusion implicitly evaluates the latent space, guiding toward PDE-compliant solutions. Moreover, the diffusion process can be viewed as transforming the deterministic PDE constraint into a stochastic differential equation (SDE), thereby enabling exploration of the solution space and providing a new perspective for bridging generative modeling with physical principles.

Building on this insight, we can empirically examine the performance of WAVEDIFFUSION with respect to the two central properties of physics-consistent latent modeling. **Completeness.** Diffusion models sample from the denoising distribution learned during training; thus, their coverage is constrained by the support of the training distribution and the noise schedule. **PDE Satisfaction.** We find that diffusion provides a close approximation to the underlying distribution $p(z)$. The dominant residual error arises from the VQGAN decoder rather than the diffusion model itself, as illustrated by the alignment observed at the beginning and end of the diffusion process in Figure 5.

## 3 EXPERIMENTS

In this section, we present the experimental evaluation of the proposed WAVEDIFFUSION framework. We conduct experiments on the full OPENFWI dataset to evaluate the model's performance in generating physically consistent seismic data and velocity maps. We assess the model's FID scores

and show its ability to improve FWI results. Then, we compare the results of training the state-of-the-art models such as BIGFWI using the jointly generated dataset against the original benchmark Jin et al. (2024). In the Appendix, we further introduce an experiment to demonstrate how the joint diffusion model compares to separately trained diffusion models in generating both two modalities.

### 3.1 DATASET AND TRAINING SETUP

In the experiments, we evaluate two different configurations for the latent diffusion models: (1) two separate VQ codebooks for the two modalities and (2) a single shared VQ codebook for both modalities. We evaluate the performance of our WAVEDIFFUSION framework using the OPENFWI dataset, a comprehensive benchmark collection comprising 10 subsets of realistic synthetic seismic data paired with subsurface velocity maps, specifically designed for FWI tasks. These subsets represent diverse geological structures, including curved velocity layers, flat velocity layers, and flat layers intersected by faults, among others, allowing for an extensive evaluation of our model's robustness and generalization capability.

Our experiments utilized all 10 subsets (Fault, Vel, and Style Families) with over 400K training data pairs to ensure a thorough assessment of WAVEDIFFUSION. We performed training on the combined dataset comprising all subsets to assess the model's ability to generalize across a variety of geological configurations and scenarios. We trained our models on 16 NVIDIA H100 GPUs. Network details and training hyperparameters used are provided in Appendix A.1.

### 3.2 EVALUATING GENERATED SAMPLES WITH FID

The Stage 1 model of our WAVEDIFFUSION framework produces coarse approximations of seismic-velocity pairs. The Stage 1 model without a diffusion process yielded an FID score of 14,207.14 for the randomly sampled $z$ vectors and their corresponding decoded velocity

Table 1: **FID scores.** The FID scores of velocity maps $v$ and seismic data $s$ for two model settings.

| Metrics \ Model | **2VQ** | **1VQ** |
|---|---|---|
| **Velocity FID** | 665.75 | 260.33 |
| **Seismic FID** | 20.94 | 5.67 |

maps and 871.31 for the decoded seismic data using an Inception-v3 model pre-trained on ImageNet Szegedy et al. (2016). A visualization example is shown in Figure 3 row 1. These high FID scores suggest that the encoder-decoder architecture, while generating plausible shapes, does not adhere closely to the true data distribution. The large disparity between seismic and velocity FID scores indicates that the generated modalities deviate more from the physical relationships governed by the wave equation as coarse approximations of the PDE solutions.

The joint diffusion model in Stage 2 is used to refine the coarse generations into physically consistent seismic-velocity pairs. We also evaluate the FID scores of both modalities generated by the joint diffusion models with different vector quantization strategies. We generated 375,000 samples for each configuration, which matched the scale of the OPENFWI training set. As shown in Table 1, among the tested configurations, the joint diffusion model with a single shared VQ layer achieved the lowest FID scores, recording 260.33 for velocity and 5.67 for seismic data, indicating a stronger latent space connection between the two modalities.

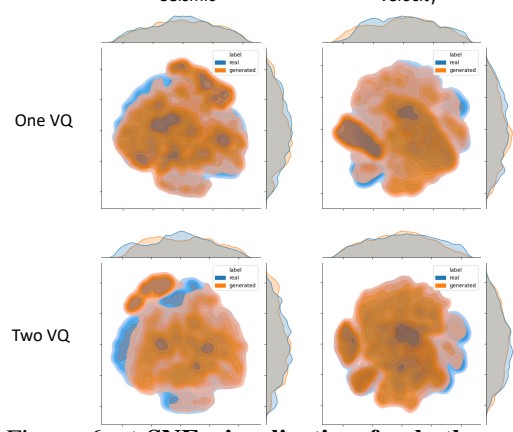

Figure 6 presents a t-SNE visualization of the feature representations extracted from Inception V3 for both real and generated data, which are used to compute the FID score. The visualization

Figure 6: **t-SNE visualization for both real (blue) and generated (orange) data.** Results of seismic data (col 1) and velocity map (col 2) for two joint diffusion models.

provides insight into the distributional alignment between real and synthesized samples in the feature space. A closer overlap between the two distributions indicates better fidelity of the generated data. This figure serves as a qualitative complement to the FID scores, illustrating how well the joint diffusion model captures the underlying data distribution.

Figure 7 visualizes results from the model with two VQs, while samples from the model with one VQ are shown in Appendix A.2. In the top row of each figure, we select samples structurally similar to FlatVel-B (FVB), where seismic exhibits perfect symmetry, and the velocity maps maintain this symmetry, demonstrating the model's ability to respect geometric constraints. Beyond reproducing individual OPENFWI subsets, we also observe cases where features from multiple datasets are fused, as seen in the ninth row, showcasing the diversity of the generated results. These results confirm that the joint diffusion model generalizes well across different data distributions, effectively capturing structural coherence and producing reliable outputs for seismic-velocity data generation.

To illustrate the potential of our approach beyond acoustic FWI, we also conducted experiments on the elastic FWI dataset, $\mathbb{E}^{\mathbf{FWI}}$ Feng et al. (2023). The results are reported in Appendix A.4.

### 3.3 DIFFUSION IMPROVES FWI

This experiment evaluates the inversion performance of our models of the two stages, specifically the encoder-decoder with two individual VQ layers (Stage 1) and its corresponding latent diffusion refinements (Stage 2). We compare these models against the BigFWI-B model, which serves as a fair baseline since its training sample volume and model parameter size approximately align with our encoder-decoder model.

For this experiment, the Stage 1 model functions as an image-to-image translation network, analogous to BigFWI-B. In contrast, the diffusion model in Stage 2 refines the Stage 1 model's latent

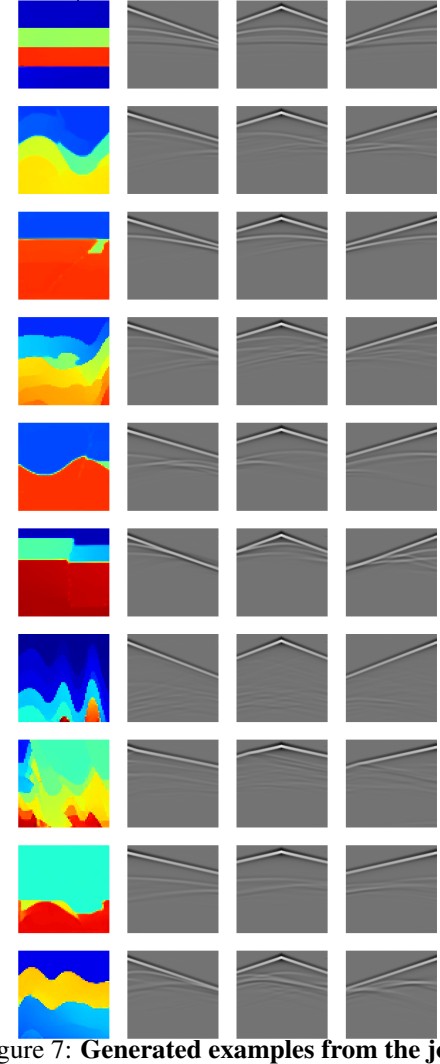

Figure 7: **Generated examples from the joint diffusion model with two VQs.**

representation using the last 10 backward denoising steps, aiming to improve reconstruction accuracy. The refinement process is performed as follows: at each denoising step, we randomly sample 100 new latent vectors $z_t$ and decode them into seismic data $\hat{s}$. Each decoded $\hat{s}$ is compared with the corresponding input seismic data $s$ by computing the L2 distance. The sample with the lowest L2 distance is selected as the current $z_t$ for the next backward denoising step. This iterative refinement continues until reaching $z_0$, at which point the final reconstructed seismic-velocity pair is obtained.

**Improvement by diffusion:** Our results, as shown in Figure 2, demonstrate that the Stage 1 model achieves inversion performance comparable to BigFWI-B across all datasets. Additionally, applying the diffusion model to refine the latent space leads to slight but consistent improvements. Visualization of examples can be found in Figure 8. These findings suggest that the latent representations learned in Stage 1 already encode meaningful physical structures, and the diffusion process further enhances them by guiding reconstructions toward physically valid solutions. For a more in-depth analysis of numerical results, we refer readers to Appendix A.5 and A.6.



Figure 8: **Inversion performance visualization.** Inverted examples from CVB (row 1), FFB (row 2), and FVB (row 3) subsets.

Table 2: **MAE** of InversionNet training on pure generated data.

| Dataset | FVA | FVB | CVA | CVB | FFA | FFB | CFA | CFB | SA | SB |
|---|---|---|---|---|---|---|---|---|---|---|
| 2VQ-Generated Data | 0.0560 | 0.1134 | 0.1001 | 0.2130 | 0.0636 | 0.1453 | 0.0706 | 0.2065 | 0.1103 | 0.1061 |
| 1VQ-Generated Data | 0.0579 | 0.1257 | 0.0952 | 0.2030 | 0.0632 | 0.1395 | 0.0730 | 0.1997 | 0.1103 | 0.1064 |
| BigFWI-B | 0.0055 | 0.0233 | 0.0343 | 0.0933 | 0.0106 | 0.0710 | 0.0167 | 0.1245 | 0.0514 | 0.0553 |

Table 3: **MAE** of InversionNet for partial real data and partial real + generated data.

| Dataset | FVA | FVB | CVA | CVB | FFA | FFB | CFA | CFB | SA | SB |
|---|---|---|---|---|---|---|---|---|---|---|
| Gen + 10%Real | **0.0192** | **0.0687** | **0.0675** | **0.1652** | **0.0262** | **0.1140** | **0.0400** | **0.1817** | **0.0862** | **0.0808** |
| Gen + 1%Real | 0.0386 | 0.1107 | 0.0846 | 0.1987 | 0.0462 | 0.1320 | 0.0588 | 0.1957 | 0.0968 | 0.0908 |
| 10%Real | 0.0328 | 0.0978 | 0.0934 | 0.2294 | 0.0453 | 0.1496 | 0.0616 | 0.2116 | 0.1121 | 0.0984 |
| 1%Real | 0.0983 | 0.2445 | 0.1507 | 0.3402 | 0.1140 | 0.1997 | 0.1339 | 0.2550 | 0.1670 | 0.1317 |

### 3.4 EVALUATING THE UTILITY OF GENERATED DATA FOR FWI

To evaluate the effectiveness of generated samples, we conduct a series of downstream experiments using image-to-image FWI models. Specifically, we assess how well the generated data can supplement the original OPENFWI dataset in low-data regimes and as an additional source for fine-tuning.

**Real data vs. generated data:** We first assess the standalone quality of the generated data by training InversionNet Wu & Lin (2019) exclusively on samples generated by WAVEDIFFUSION, using a training set that matches the scale of the original OPENFWI dataset. The model is then evaluated on the OPENFWI test set, and the results are compared with the state-of-the-art BigFWI-B Jin et al. (2024) baseline. As shown in Table 2, models trained purely on generated samples underperform compared to those trained on real data, indicating a gap in fidelity despite structural consistency.

**Improvement by combining limited real data with generated data:** We further examine whether generated data can improve performance when real training data is limited. Focusing on samples generated by the one-VQ variant of our model, we compare two settings: where generated data is combined with a small portion (10% or 1%) of real data, and $n$%Real, where models are trained exclusively on the same small subset of real data. Results in Table 3 show that combining a small fraction of real data with generated samples significantly boosts model performance compared to using either source alone. Gen+10%Real substantially outperforms 10%Real alone, with performance approaching the BigFWI-B. Similarly, Gen+1%Real consistently surpasses both 1%Real, especially on more complex datasets, and the model trained purely on generated data (Table 2). These results highlight two key findings: generated data is beneficial when used alongside even a small amount of real data, and such combinations are markedly more effective than training on limited real or synthetic data alone. This validates that the generated samples, while not perfectly matching, are distributed closely enough to the real data to be a supplementary source.

Figure 9 illustrates prediction visualizations, while additional results are provided in Appendix A.7. Overall, these results demonstrate that while generated samples can effectively supplement small datasets, the inclusion of even a small portion of real data is crucial for achieving optimal results.

**Providing additional data with challenging samples:** In addition to low-data scenarios, we evaluate the benefit of using generated samples as additional training data to fine-tune a pretrained model. Specifically, we select the 65K most challenging generated samples, on which BigFWI-B performs worst, and partition them into 55K for training and 10K for testing. We then fine-tune BigFWI-B for 20 additional epochs on the combined dataset (original OPEN-FWI + generated challenging samples). For a

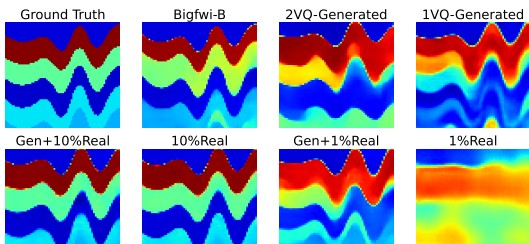

Figure 9: **InversionNet performance visualization.** Examples from CVB subset.

fair comparison, we compare with a model fine-tuned only on the original OPENFWI training set for the same number of epochs. As shown in Table 4, fine-tuning with generated challenging samples yields consistent improvements across all ten datasets in OPENFWI, along with a significant boost in performance on the generated test samples. This suggests that WAVEDIFFUSION not only produces physically meaningful samples but also exposes hard modes that enhance model performance.

Table 4: **MAE** of InversionNet for fine-tuning on the combined dataset versus OpenFWI-only. LDM denotes performance on 1K generated challenging samples.

| Dataset | FVA | FVB | CVA | CVB | FFA | FFB | CFA | CFB | SA | SB | LDM |
|---|---|---|---|---|---|---|---|---|---|---|---|
| Combined data | **0.0062** | 0.0220 | **0.0338** | **0.0927** | **0.0097** | **0.0716** | **0.0160** | **0.1256** | **0.0512** | **0.0548** | **0.1900** |
| OpenFWI only | 0.0064 | 0.0220 | 0.0339 | 0.0933 | 0.0102 | 0.0718 | 0.0163 | 0.1259 | 0.0513 | 0.0549 | 0.2120 |

## 4 RELATED WORKS

**Data-driven approaches to FWI:** Recent learning-based methods bypass iterative solvers by training neural networks to map seismic data to velocity models directly. Encoder-decoder architectures such as *InversionNet* Wu & Lin (2019) and *VelocityGAN* Zhang et al. (2019) reduce computational cost by learning implicit mappings between the two modalities. With higher efficiency than physics-based FWI, these methods treat FWI as an image-to-image translation task Richardson (2018); Zhu et al. (2019). More recently, neural operators Li et al. (2020; 2023) provide a flexible alternative for predicting seismic data from velocity models, with strong capacity for modality transformation. In Appendix A.9, we review traditional physics-driven FWI methods.

**Generative models in FWI:** Generative models, particularly Generative Adversarial Networks (GANs) and their variants, have emerged as alternatives to traditional CNN-based methods for FWI. These models aim to learn the latent representations of seismic data and velocity models, enabling the generation of synthetic training data or even direct inversion Goodfellow et al. (2020). Vector Quantized GANs (VQGANs) Esser et al. (2021), in particular, have been explored for their ability to generate high-quality modalities, such as images, audios, videos, etc. Such models can be tuned for imaging one physical modality (e.g., velocity) given another (e.g., seismic) Zhang et al. (2019).

Diffusion models have also been applied to FWI Wang et al. (2023a), who used them to generate prior distributions for plausible velocity models and incorporated with a regularization process through a Plug-and-Play (PnP)-style framework. However, their approach treats seismic data and velocity maps separately and relies on explicit forward modeling. Recent work on Latent Diffusion Models (LDMs) Ho et al. (2020); Dhariwal & Nichol (2021); Rombach et al. (2022) refines latent representations through a diffusion process. While effective for generating realistic single-modality data, these models are difficult to produce multiple physical consistency modalities jointly.

## 5 LIMITATION

While this work introduces a novel generative perspective on seismic and velocity data modeling, several limitations remain. First, we provide no formal guarantees or theoretical characterization of PDE satisfaction. Our contribution is to highlight and empirically quantify the emergent property and tendency of improved physical adherence in the generated seismic-velocity pairs from the joint latent diffusion. Second, despite the diffusion process promoting physical consistency, the PDE satisfaction is not perfect. This is a common limitation of data-driven methods that approximate complex physical systems. Finally, although we conducted preliminary experiments on the elastic FWI setting, our study remains primarily focused on the acoustic case. A more systematic investigation of more wave physics or other inverse problems remains an important direction for future research.

## 6 CONCLUSION

In this work, we revisit the relationship of seismic and velocity data from a generative perspective and propose WAVEDIFFUSION, a diffusion-based framework for physics-consistent latent modeling. We demonstrate that standard image-to-image translation FWI models do not ensure physical consistency in their learned latent representations. To address this, we employ a joint diffusion model in the shared latent space that learns to evaluate deviations from the governing PDE, progressively transforming arbitrary latent points into physically consistent solutions, ensuring that generated seismic-velocity pairs naturally satisfy the governing PDE. Additionally, our experiments show that the diffusion model can improve the performance of FWI tasks, and the generated data can serve as a supplement to the training set for data-driven FWI models. Our findings provide a new perspective on bridging generative modeling with physics-based problem-solving, paving the way for diffusion models to enhance scientific discovery and computational physics applications.

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

# A APPENDIX

## A.1 NETWORK DETAILS AND TRAINING HYPERPARAMETERS

In this appendix section, we provide details on the network architectures and training hyperparameters used for the encoder-decoder FWI model and joint diffusion models in our experiments.

### A.1.1 ENCODER-DECODER MODEL WITH SHARED LATENT SPACE AND VECTOR QUANTIZATION

**Shared Latent Space Construction:** In the first stage of the WAVEDIFFUSION framework, we build a shared latent space using a VQ-VAE architecture that comprises one encoder and two decoders. The encoder $E$ maps the input seismic data $s$ into a continuous latent representation $z_{\text{cont}}$, which is then discretized by a vector quantization (VQ) layer into $z$. Two decoders, $D_s$ and $D_v$, are jointly trained to reconstruct the seismic data ($\hat{s}$) and the corresponding velocity maps ($\hat{v}$) from the same latent code $z$. This joint training ensures that the latent space captures the intrinsic dependencies between the two modalities, thereby enabling paired generation.

To clarify the training procedure, Algorithm 1 outlines our approach.

---

**Algorithm 1** Training the Shared Latent Space in WAVEDIFFUSION

---

**Require:** Training pairs $\{(s, v)\}$
1: **Initialize:** Encoder $E$, decoders $D_s$ and $D_v$, vector quantizer (VQ), and discriminator (for perceptual loss).
2: **for** each training iteration **do**
3:     Compute continuous latent code: $z_{\text{cont}} = E(s)$.
4:     Quantize latent code: $z = \text{VQ}(z_{\text{cont}})$.
5:     Reconstruct seismic data: $\hat{s} = D_s(z)$.
6:     Reconstruct velocity map: $\hat{v} = D_v(z)$.
7:     Compute reconstruction losses:

$$\mathcal{L}_s = \|s - \hat{s}\|_1, \quad \mathcal{L}_v = \|v - \hat{v}\|_1.$$

8:     Compute perceptual loss $\mathcal{L}_{\text{perc}}$ (using a discriminator or LPIPS loss).
9:     Form the total loss:

$$\mathcal{L} = \mathcal{L}_s + \mathcal{L}_v + \mathcal{L}_{\text{perc}}.$$

10:     Backpropagate $\mathcal{L}$ and update $E$, $D_s$, $D_v$, and the VQ layer.
11: **end for**

---

**Vector Quantization:** The VQ layer discretizes the continuous latent vectors into a finite codebook (with an embedding dimension of 32 and a codebook size of 8192). This discretization helps capture structured patterns and long-range dependencies between seismic data and velocity maps.

**Architectural Details:** The encoder and decoders are implemented as convolutional encoder-decoder networks using ResNet blocks He et al. (2016). For velocity maps, the channel multipliers are set to $[1, 2, 2, 4, 4]$ with an output resolution of 64. For seismic data, the multipliers are $[1, 2, 2, 4, 4, 4, 4, 8, 8]$ with resolutions $[1024, 64]$. The latent feature map $z$ has dimensions $[16, 16]$, and 3 residual blocks are used. The model is trained with a base learning rate of $4.5 \times 10^{-4}$ and employs a perceptual loss combined with an adversarial loss. The discriminator begins training at step 50001 with weights of 0.5 for both the discriminator and perceptual loss components.

### A.1.2 JOINT DIFFUSION MODEL

The Joint Diffusion model is based on the `LatentDiffusion` architecture. The backbone network in the Joint Diffusion model is a UNet-based architecture. The UNet takes 32 input and output channels, and the model channels are set to 128. The attention resolutions are [1, 2, 4, 4], corresponding to spatial resolutions of 32, 16, 8, and 4. The model uses 2 residual blocks and channel multipliers of

[1, 2, 2, 4, 4]. It also employs 8 attention heads with scale-shift normalization enabled and residual blocks that support upsampling and downsampling. The model is trained with a base learning rate of $5.0 \times 10^{-5}$ and uses 1000 diffusion timesteps. The loss function applied is $L_1$. The diffusion process is configured with a linear noise schedule, starting from 0.0015 and ending at 0.0155.

A LambdaLinearScheduler is used to control the learning rate, with 10000 warmup steps. The initial learning rate is set to $1.0 \times 10^{-6}$, which increases to a maximum of 1.0 over the course of training.

### A.1.3 TRAINING HYPERPARAMETERS

Both the encoder-decoder model and joint diffusion models were trained using the Adam optimizer, with $\beta_1 = 0.9$ and $\beta_2 = 0.999$. The models were trained with a batch size of 16 for 500 epochs. The learning rate follows an exponential decay schedule with a decay rate of 0.98. Gradient clipping was applied with a threshold of 1.0. Early stopping was implemented when the validation loss plateaued for 10 consecutive epochs.

We trained our models on 128 NVIDIA GH200 GPUs. Training required approximately 8000 GPU hours for the first-stage encoder-decoder model and 12000 GPU hours for the joint diffusion model.

Seismic data and velocity models were resized from [5,70,1000]/[1,70,70] to [3,64,1024]/[1,64,64] (channel, height, depth) for consistency with our architecture. Log transform is performed for seismic data. Both were normalized to [-1,1] to ensure compatibility and stability.

### A.2 JOINT GENERATION EXAMPLES

We illustrate generated samples from the model with one VQ in Figure 10. In the top row, we present samples structurally similar to FlatVel-B (FVB), where the seismic inputs exhibit perfect symmetry along a central vertical plane. The corresponding generated velocity maps preserve this symmetry, demonstrating the model's ability to respect the geometric constraints of the input data.

The model also generates samples that fuse structures from multiple datasets, as observed in the second row of Figure 10. These examples showcase that the model effectively handles a wide range of input patterns while maintaining physical and structural coherence.

### A.3 EVALUATING LATENT DISTRIBUTION SAMPLING: RANDOM SAMPLING AND INTERPOLATION

We conducted two experiments to address the concern that poor reconstructions result merely from latent distribution mismatch.

1. **Random Sampling Experiment:** We first normalized the training latent vectors (300K samples) to a Gaussian (zero mean, std=1) and then sampled 5000 new points from this distribution. The decoded velocity maps from these new points show merged features from the training data. However, the corresponding reconstructed seismic data ($\hat{s}$) exhibit significantly larger differences from the ground truth seismic data $s$ (obtained via finite-difference simulation on the decoded velocity maps). This confirms that poor reconstructions are not merely due to sampling from a mismatched distribution. Visualizations can be found at Figure 11.

2. **Interpolation Experiment:** We interpolated between two training latent points (A and B, 5000 pairs in total) using five different weights (i.e., $0.1A + 0.9B$, $0.3A + 0.7B$, $0.5A + 0.5B$, $0.7A + 0.3B$, and $0.9A + 0.1B$). While the decoded velocity maps smoothly merge features of A and B, the resulting seismic data differences ($\hat{s} - s$) remain large. This indicates that the auto-encoder's latent space does not inherently support paired generation, underscoring the necessity of the diffusion process to refine the shared latent space for PDE-consistent outputs. Visualizations can be found at Figure 12.

To clarify, the two experiments above do not contain any diffusion process, and are only performed using the first stage encoder-decoder network. The results are summarized in Table 5:

These results demonstrate that encoder-decoder sampling (both random and interpolation) yields relatively high MSE values, which indicates most latent space points do not correspond to valid PDE

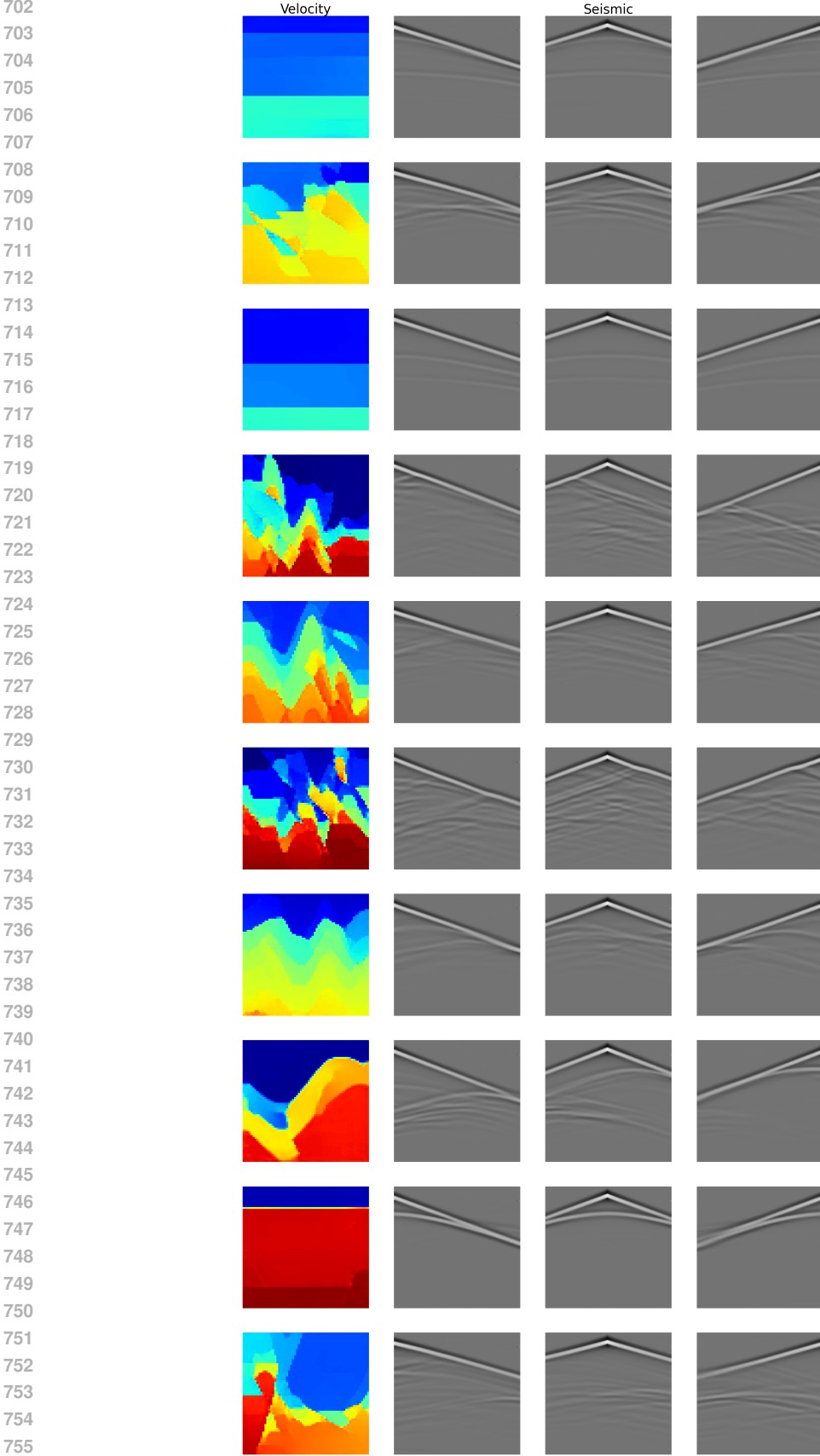

Figure 10: **Generated Examples from the joint diffusion model with one VQ.**

solutions. The diffusion process significantly reduces the MSE, confirming that the diffusion process guides latent codes toward outputs that satisfy the PDE much more effectively.

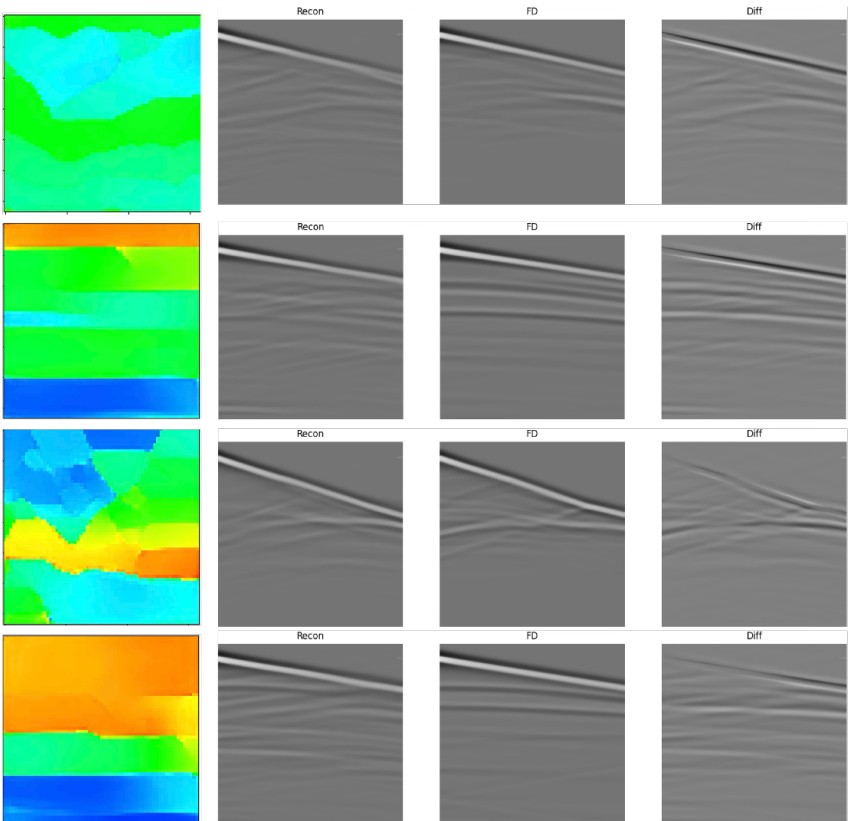

Figure 11: **Random Sampling Experiment.**

Table 5: **Evaluating latent distribution sampling,** comparing diffusion with random Sampling and interpolation.

| Methods | Seismic Averaged MSE |
|---|---|
| Random Sampling | 0.0082 |
| Interpolation Sampling | 0.0065 |
| Diffusion Denoising | **0.0038** |

## A.4 EXTENSION TO ELASTIC FWI

We further evaluated the generality of our framework by conducting experiments on one sub-dataset (FFA) from the $\mathbb{E}^{\mathbf{FWI}}$ dataset Feng et al. (2023). To adapt our network to the elastic setting, we stacked the two seismic components ($x$ and $z$) along the channel dimension, and likewise stacked the $P$- and $S$-wave velocity maps. Using the trained model, we generated 6K samples and report the FID scores across all four modalities in Table 6.

| | FID (VP) | FID (VS) | FID (Data_x) | FID (Data_z) |
|---|---|---|---|---|
| Value | 195.91 | 301.77 | 3.40 | 1.15 |

Table 6: FID scores for elastic FWI across different modalities.

These results are comparable to those obtained in the acoustic FWI experiments, suggesting that our framework extends naturally to the elastic case. In terms of PDE satisfaction, we observed a

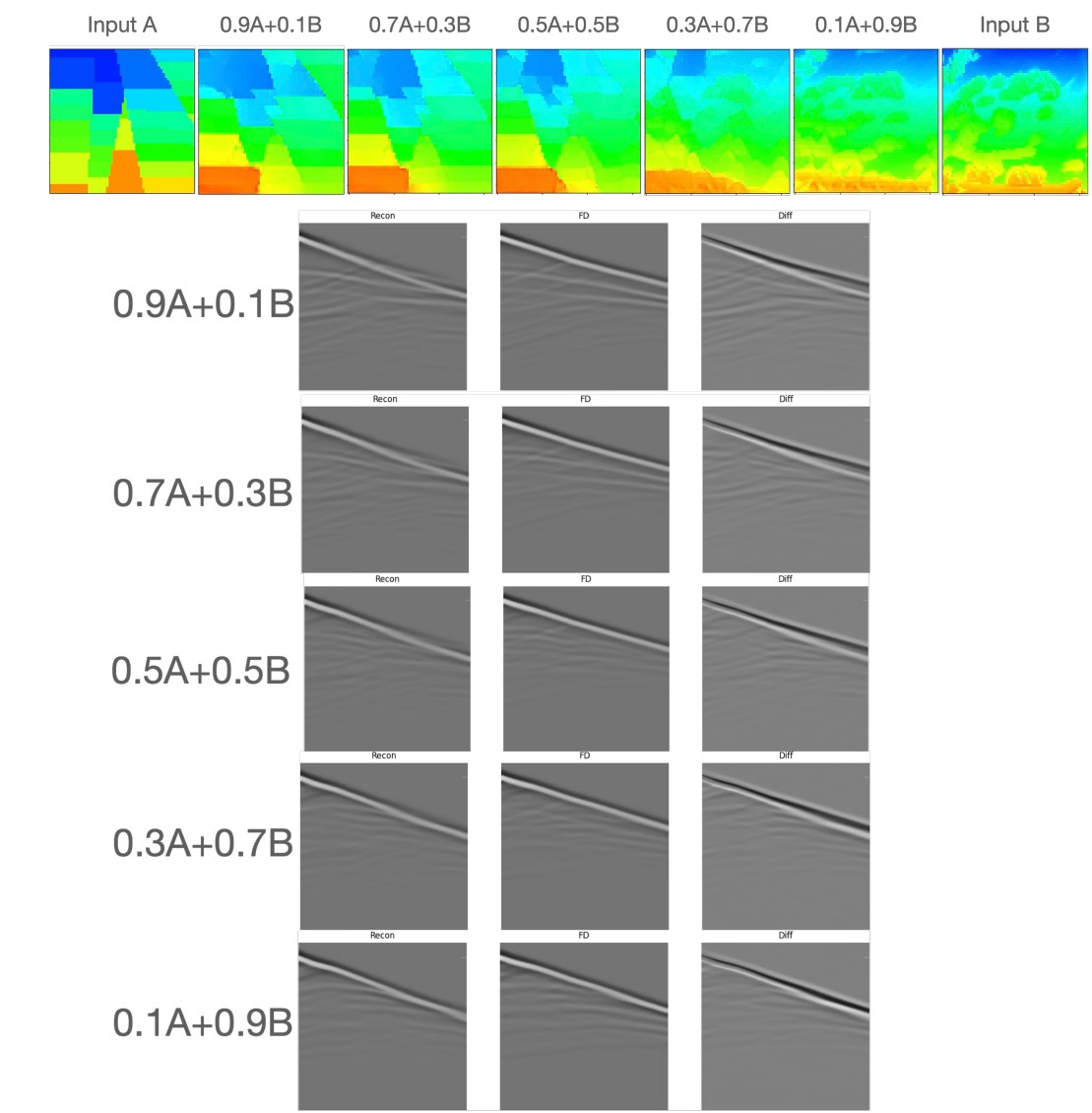

Figure 12: **Interpolation Experiment.**

similar denoising trend: the average $L_2$ deviation for the $x$-component of seismic data decreased from $7.98 \times 10^{-4}$ to $7.80 \times 10^{-4}$ during denoising, while for the $z$-component it decreased from $1.8 \times 10^{-5}$ to $0.8 \times 10^{-5}$. Notably, these deviations are even lower than in the acoustic setting, which we attribute to training on a single sub-dataset, where more consistent data patterns make learning easier.

A.5 STATISTICAL COMPARISON OF OUR MODELS AND BIGFWI-B

We present the detailed statistical comparison in Table 9 of the BigFWI-B model and our first-stage encoder-decoder models and the corresponding diffusion models on the inversion tasks for the velocity maps across all the datasets of OPENFWI.

A.6 ABLATION OF LATENT REFINEMENT VIA DIFFUSION MODEL

In addition to our initial experiment, we conducted a further analysis as follows: We randomly sampled 1,000 latent vectors from the Gaussian distribution centered around the latent position corresponding to the target seismic data (from the Style-B dataset) with a slight oscillation ($5\%$) from the target seismic data encoded latent point. Next, for each sample, we decoded the seismic data ($\hat{s}$)

and computed the MSE loss with respect to the target seismic data $s$. We then selected the latent point with the lowest MSE loss and evaluated the paired velocity map using RMSE, MAE, and SSIM metrics.

The results are summarized in Table 7. An example comparison can be found at Figure 13. These results indicate that the encoder-decoder random sampling cannot improve over the baseline, and the diffusion model FWI consistently achieves lower MAE and RMSE, as well as a higher SSIM, demonstrating an improvement by the diffusion process. The random sampling around the baseline latent point by the encoder-decoder network only even decreases the performance. This confirms that the diffusion process effectively refines the latent representation, yielding samples that better satisfy the governing PDE and thus improve FWI performance.

Table 7: **Ablation of Latent Refinement via Diffusion model,** comparing with ocillation sampling around baseline on the SB dataset.

| Methods | MAE | RMSE | SSIM |
|---|---|---|---|
| Starting Baseline | 0.0574 | 0.0882 | 0.9026 |
| Oscillation Sampling around Baseline | 0.1199 | 0.1582 | 0.7722 |
| Diffusion Model FWI | 0.0423 | 0.0873 | 0.9069 |

Encoder-decoder predicted     Diffusion model predicted     Ground Truth

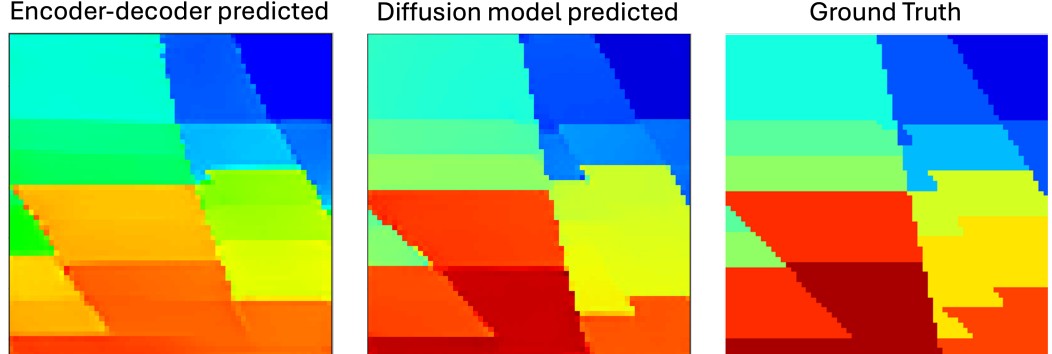

Figure 13: **Visualization of Latent Refinement via Diffusion model.**

### A.7 INVERSIONNET RESULTS VISULIZATION

In Figure 15, we illustrate more visualizations of InversinNet Results trained on different training data settings.

### A.8 SEPARATE VS. JOINT DIFFUSION

We compare the joint diffusion model with separate diffusion models, where seismic data and velocity maps are generated independently. In the separate models, the latent space constructed by the single-branch encoder-decoder lacks a shared representation. Both approaches are trained on the CVB subset, with results summarized in Table 8 and Figure 14.

The joint diffusion model consistently outperforms the separate models in FID scores. For the CVB dataset, joint diffusion achieves FID scores of 30.66 (seismic) and 186.86 (velocity), compared to 131.48 and 411.40, respectively, for the separate models. This highlights the superior quality of the joint diffusion outputs.

Beyond visual quality, the joint model enforces physical consistency with the wave equation, which the separate models fail to achieve. As shown in Figure 14, separate models exhibit significant deviations from the governing PDE, while the joint diffusion model generates seismic-velocity pairs that are both visually realistic and physically valid. This underscores the effectiveness of the WAVEDIFFUSION framework in maintaining fidelity and physical correctness.

Table 8: **FID score comparison between separate and joint generations.** Evaluations on seismic data $s$ and velocity maps $v$ on the CVB dataset.

| Modality | LDM Setup | FID |
|----------|-----------|-----|
| Velocity | Joint | 186.86 |
| | Separate | 411.40 |
| Seismic | Joint | 30.66 |
| | Separate | 131.48 |

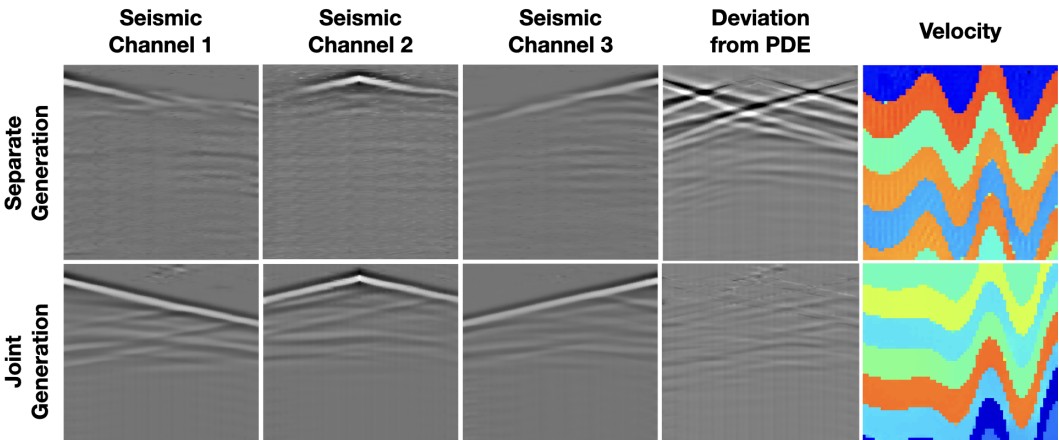

Figure 14: **Visualization of separate vs. joint diffusion.** Row 1: Separate models; Row 2: Joint model. Column 4 shows deviations from the governing PDE.

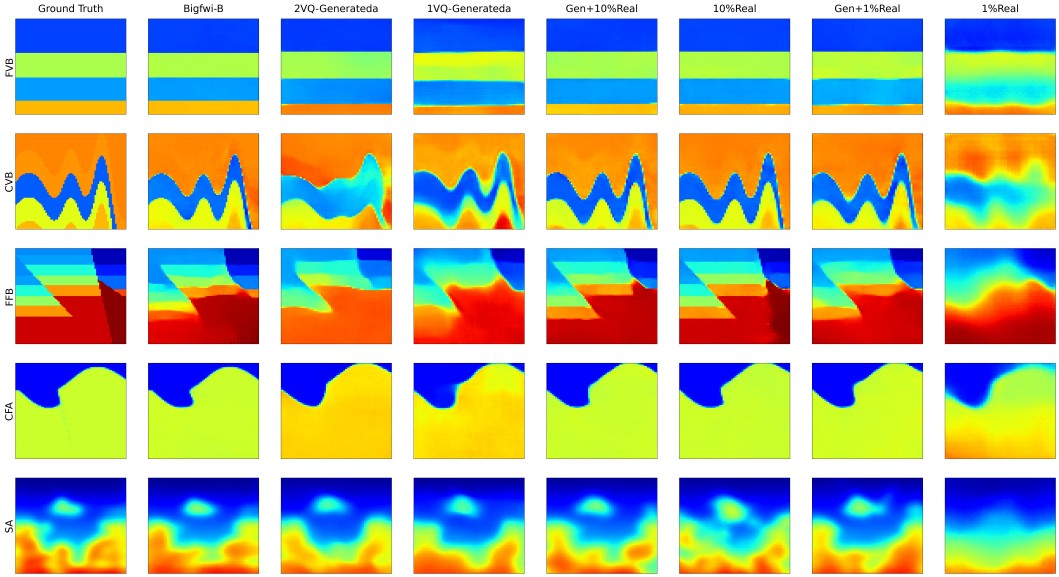

Figure 15: **InversionNet performance visualization.** The predictions on the FVB, CVB, FFB, CFA, and SA subsets.

### A.9 RELATED WORKS OF TRADITIONAL PHYSICS-BASED FWI

Traditional FWI methods aim to reconstruct subsurface velocity models by iteratively minimizing the difference between observed and simulated seismic data, typically using gradient-based optimization methods. The key challenge lies in solving the wave equation, which governs wave propagation through the Earth. While effective, these methods are computationally expensive and sensitive to factors such as the quality of the initial velocity model, noise in the data, and cycle-skipping issues—where the inversion algorithm converges to incorrect solutions due to poor starting models or insufficient low-frequency data Tarantola (1984); Virieux & Operto (2009). Techniques such

Table 9: **Performance Comparison of Stage 1 Encoder-Decoders and Stage 2 Diffusion Models on OpenFWI Datasets.** Metrics include Mean Absolute Error (MAE), Root Mean Squared Error (RMSE), and Structural Similarity Index (SSIM). Lower values for MAE and RMSE indicate better performance, while higher SSIM values indicate better structural similarity.

| Dataset | Metric | Model | | | | |
|---|---|---|---|---|---|---|
| | | BigFWI-B | 2VQ | 1VQ | 2VQ LDM | 1VQ LDM |
| FlatVel_A | MAE | **0.0055** | 0.0097 | 0.0197 | 0.0087 | 0.0198 |
| | RMSE | 0.0130 | 0.0133 | 0.0254 | **0.0113** | 0.0222 |
| | SSIM | 0.9943 | 0.9974 | 0.9943 | **0.9989** | 0.9958 |
| FlatVel_B | MAE | 0.0233 | 0.0203 | 0.0396 | **0.0195** | 0.0402 |
| | RMSE | 0.0696 | **0.0273** | 0.0619 | 0.0285 | 0.0560 |
| | SSIM | 0.9658 | **0.9952** | 0.9746 | 0.9936 | 0.9776 |
| CurveVel_A | MAE | 0.0343 | 0.0281 | 0.0383 | **0.0251** | 0.0315 |
| | RMSE | 0.0798 | 0.0652 | 0.0603 | 0.0604 | **0.0585** |
| | SSIM | 0.9027 | 0.9282 | 0.9401 | 0.9384 | **0.9451** |
| CurveVel_B | MAE | 0.0933 | 0.0658 | 0.0794 | **0.0640** | 0.0733 |
| | RMSE | 0.2154 | 0.1610 | 0.1523 | 0.1598 | **0.1519** |
| | SSIM | 0.7808 | 0.8541 | 0.8590 | 0.8556 | **0.8633** |
| FlatFault_A | MAE | **0.0106** | 0.0186 | 0.0197 | 0.0148 | 0.0186 |
| | RMSE | 0.0286 | **0.0231** | 0.0279 | 0.0237 | 0.0262 |
| | SSIM | 0.9871 | 0.9893 | 0.9871 | **0.9901** | 0.9893 |
| FlatFault_B | MAE | 0.0710 | 0.0574 | 0.0455 | **0.0423** | 0.0444 |
| | RMSE | 0.1321 | 0.0882 | 0.0862 | 0.0873 | **0.0859** |
| | SSIM | 0.8027 | 0.9026 | 0.9073 | 0.9069 | **0.9084** |
| CurveFault_A | MAE | **0.0167** | 0.0195 | 0.0236 | 0.0182 | 0.0244 |
| | RMSE | 0.0474 | 0.0388 | 0.0435 | **0.0366** | 0.0402 |
| | SSIM | 0.9712 | 0.9751 | 0.9603 | **0.9794** | 0.9671 |
| CurveFault_B | MAE | 0.1245 | 0.1088 | 0.1075 | 0.0905 | **0.0891** |
| | RMSE | 0.2027 | 0.1653 | 0.1640 | **0.1620** | 0.1624 |
| | SSIM | 0.6781 | 0.7490 | 0.7507 | 0.7548 | **0.7551** |
| Style_A | MAE | 0.0514 | 0.0553 | 0.0575 | **0.0462** | 0.0495 |
| | RMSE | 0.0868 | 0.0730 | 0.0767 | **0.0694** | 0.0738 |
| | SSIM | 0.9125 | 0.9334 | 0.9313 | **0.9384** | 0.9359 |
| Style_B | MAE | **0.0553** | 0.0626 | 0.0637 | 0.0637 | 0.0630 |
| | RMSE | **0.0876** | 0.0969 | 0.0947 | 0.0960 | 0.0944 |
| | SSIM | **0.7567** | 0.7289 | 0.7368 | 0.7320 | 0.7381 |

as adaptive waveform inversion Warner & Guasch (2016) and multiscale FWI Bunks et al. (1995) have been developed to reduce the risk of cycle-skipping and improve convergence by progressively introducing higher-frequency data. These techniques frame FWI as a conditional generation problem, relying on physical equations as the computational foundation Virieux et al. (2017); Warner & Guasch (2016).

## A.10 USAGE OF LARGE LANGUAGE MODELS (LLMs)

During the preparation of this manuscript, we used a large language model (LLM) to assist with language polishing, structural refinement, and presentation clarity. The LLM provided feedback on phrasing, grammar, and flow, suggested alternatives to reduce redundancy, and generated LaTeX formatting for tables and equations. All scientific ideas, experiments, analyses, and conclusions were conceived, designed, and carried out entirely by the authors.

