# OpenReview forum: "WaveDiffusion: Joint Latent Diffusion for Physically Consistent Seismic and Velocity Generation"
_ICLR.cc/2026/Conference — Submitted to ICLR 2026_

### Official Review · Reviewer_7gNb · 2025-10-27

**Soundness:** 3
**Presentation:** 2
**Contribution:** 2
**Rating:** 4
**Confidence:** 4

**Summary:**

The paper introduces WaveDiffusion, a diffusion-based framework for FWI that jointly models seismic waveforms and subsurface velocity maps through a shared latent diffusion process. Unlike traditional inversion methods that treat FWI as a deterministic image-to-image translation, WaveDiffusion learns a latent space where both modalities satisfy the acoustic wave equation, ensuring physical consistency between generated pairs. The model uses a two-stage design: first, and encoder-decoder establishes a shared latent representation; second, a joint diffusion model refines latent samples toward PDE-consistent solutions without explicit physical constraints. Experiments on the OpenFWI benchmark show that the approach yields high-fidelity, diverse, and physically valid seismic-velocity pairs, improving upon state-of-the-art models like BigFWI-B in both reconstruction accuracy and generalization. Furthermore, the generated data enhances the training of supervised inverse models, demonstrating that latent diffusion inherently encodes physics-aware structure, providing a promising bridge between generative AI and scientific modeling.

**Strengths:**

1. The paper provides comprehensive experimental validation, comparing the proposed approach across multiple settings, and visualizing the progressive process, demonstrating its robustness and consistency.

2. The method effectively refines latent representations through a progressive diffusion process, offering new insights into how latent diffusion can enforce physical consistency in high-dimensional scientific data.

**Weaknesses:**

1. The approach appears to require retraining when observed geometries differ from those used in training, which may limit its applicability in real-world seismic scenarios with varying acquisition setups.

2. As acknowledged by the authors, the method lacks a theoretical guarantee for PDE satisfaction or convergence to physically consistent solutions. To strengthen the work, the authors should include comparisons or at least a discussion with related diffusion-based inverse modeling studies, such as [1], which address theoretical and empirical aspects of physics-constrained diffusion.

[1] H. Zheng, et. al., InverseBench: Benchmarking Plug-and-Play Diffusion Priors for Inverse Problems in Physical Sciences, ICLR, 2025.

**Questions:**

1. While the framework performs well on FWI, could the authors clarify why the study focuses exclusively on seismic inversion rather than extending the approach to other physics-based forward/inverse problems?

---

### Official Review · Reviewer_zkSB · 2025-10-31

**Soundness:** 3
**Presentation:** 1
**Contribution:** 2
**Rating:** 2
**Confidence:** 3

**Summary:**

WaveDiffusion is a way to improve the joint distribution of seismic and velocity images that is learned by autoencoder-style image-to-image solvers (which map from seismic images to velocity images). Specifically, a diffusion model is trained in the latent space. The authors find that sampling from the latent diffusion model gives more physically-consistent (in terms of PDE agreement) samples than sampling from the original learned latent space. In their experiments, the authors show that WaveDiffusion can improve the performance of BigFWI-B, a state-of-the-art network for full waveform inversion. They also show that synthetic data generated with WaveDiffusion can help improve the performance of FWI models on challenging datasets.

**Strengths:**

* The proposed method is sound and appears to generally give more physically consistent samples than the original latent space of a FWI network.

**Weaknesses:**

* The authors only compare to BigFWI-B and are missing a diffusion-model-based baseline. See InverseBench (Zheng et al. ICLR 2025) for a comparison of state-of-the-art diffusion inverse solvers on the FWI task. According to their findings, DiffPIR (Zhu et al. ICCV 2023) would give the most accurate results. It is important to compare to a diffusion-based baseline since it is hard to tell whether WaveDiffusion is the best way to utilize a diffusion model for this problem.
* I struggle to understand one of the main motivations of the paper, which is that “most latent points do not correspond to PDE solutions.” First, it would be good to quantify this claim by sampling many times from the original distribution and reporting the percentage of samples that satisfy the PDE solution (within some error tolerance). Second, the original latent distribution was specifically trained for the task of inversion, so I wouldn’t expect it to perform particularly well for unconditional generation anyways. Perhaps I am missing something about why we should expect the inversion network to give a good generative distribution.
* According to Figure 2, there are some datasets where WaveDiffusion doesn’t improve upon BigFWI-B. It would be good for the authors to discuss why some of these datasets might be more challenging.
* In general a lot of the figures could use more explanation. For Figure 3, I guess the point is that the stage 2 velocity is better than the stage 1 velocity, but it’s hard to tell that by just looking at the image. For Figure 6, showing the entire t-SNE plot does not seem to be necessary, unless the authors can comment on any interesting features of the t-SNE plots themselves.
* I do not understand the point of Figure 5. It doesn’t seem to be either surprising or concerning that deviation from the PDE would be large at high noise levels. The important thing is whether the PDE deviation is low at a near-zero noise level.
* The authors claim that diffusion models implicitly guide samples to be PDE-compliant. It would be nice if they could offer some explanation as to why this might be the case.

**Questions:**

* Please comment on why we should expect the latent distribution of an inversion network to perform well for unconditional generation.
* Please compare your method to another plug-and-play diffusion-based inverse solver (see InverseBench for methods that were successfully applied to FWI).
* Please comment on why the experiment on PDE deviation vs. diffusion noise level is useful.

---

### Official Review · Reviewer_mNJb · 2025-10-31

**Soundness:** 2
**Presentation:** 3
**Contribution:** 1
**Rating:** 2
**Confidence:** 4

**Summary:**

This paper proposes WaveDiffusion, a joint latent diffusion framework for generating physically consistent seismic-velocity pairs in Full Waveform Inversion. While the empirical observation that diffusion refines latent codes toward PDE-consistent solutions is interesting, the work suffers from unsubstantiated claims about physical consistency, lacks ML novelty beyond applying standard latent diffusion, uses unrealistically small synthetic datasets, and makes conclusions that overreach what the results actually demonstrate.

**Strengths:**

Writing is mostly clear

**Weaknesses:**

- The paper repeatedly claims generated pairs "satisfy the governing PDE" but only measures L2 distance between decoded seismics and finite-difference solutions, which remains non-trivial (0.002) even after diffusion. No rigorous definition of what constitutes "satisfying" the PDE is provided, and the threshold ε in the validity criterion is never specified or justified.

- The approach is a straightforward application of VQ-VAE + latent diffusion (Rombach et al. 2022) to paired seismic-velocity data. The only architectural modification is adding a second decoder branch, which is a trivial extension. No new diffusion techniques, training objectives, or theoretical insights are contributed.

- All experiments use synthetic OpenFWI data (400K samples) with simple geological structures and perfect acoustic assumptions. Real seismic data involves noise, complex elastic wave propagation, acquisition artifacts, and missing low frequencies—none of which are addressed. The elastic FWI experiment (Appendix A.4) uses only a single subdataset (6K samples), making generalization claims unjustified.

- The claim that diffusion "implicitly evaluates the latent space" and "naturally enforces physical consistency" is not supported. The model is trained purely on data pairs without any physics loss or PDE constraints, so any PDE adherence is correlation from data distribution, not causal enforcement. The paper conflates sampling from a learned data distribution with satisfying physical laws.

- The BigFWI-B baseline is described as having "approximately aligned" training volume and parameters (24M vs 19M), but details on training procedures, data augmentation, and hyperparameter tuning for fair comparison are missing. The Stage 2 diffusion refinement for FWI (Section 3.3) uses a cherry-picking procedure (selecting best of 100 samples per denoising step) that is computationally prohibitive and not practical.

- Table 2 shows models trained purely on generated data significantly underperform those trained on real data (e.g., CVB: 0.2030 vs 0.0933 MAE), indicating the generated samples are far from matching real data distribution despite low FID scores. This contradicts claims about high-fidelity generation and raises questions about what FID actually measures in this domain.

- The comparison between 1VQ and 2VQ configurations (Table 1) lacks explanation of why shared codebooks improve physical consistency. The interpolation/random sampling experiments (A.3) are relegated to the appendix and don't clearly separate the effects of distribution mismatch vs. inherent decoder limitations. No analysis of what latent dimensions encode physical vs. spurious correlations.

- The paper states prior work focuses on "inverse problems" while this work takes a "generative perspective," but doesn't articulate why jointly generating both modalities is a meaningful task. Real FWI applications need to invert seismic-velocity, not generate paired synthetic data. The physics-consistent latent modeling task (Section 2.1) is formalized but its practical utility beyond data augmentation is unclear.

- The two-stage training requires first training a VQ-VAE encoder-decoder (8000 GPU hours on 128 GH200 GPUs) then training a diffusion model (12000 GPU hours), totaling ~20,000 GPU hours for 400K samples. Scaling to realistic 3D seismic volumes (currently 2D cross-sections) or real field data would be computationally prohibitive, yet no discussion of computational costs, inference time, or scalability limitations is provided.

- The paper uses imprecise language (e.g., "emergent property," "implicitly biases," "sparse subset") without quantification. Figure 1's caption claims diffusion maps "non-solution points (gray squares in valleys) to valid solutions (colored stars at peaks)" but this visualization is metaphorical and not derived from actual analysis. The completeness and PDE satisfaction properties (Section 2.1) are introduced but never rigorously evaluated in experiments.

**Questions:**

See above

---

### Meta-Review · Area_Chair_r5Zz · 2025-12-14

**Summary:**

This paper introduces WaveDiffusion, which trains a standard latent diffusion model over the joint latent space of seismic observations and velocity models produced by an autoencoder-style FWI pipeline, and reports improved PDE-consistency and downstream performance. Reviewers acknowledge the empirical finding that diffusion sampling can “refine” latent codes toward solutions with lower wave-equation residuals as interesting. However, the submission’s central claims about physical consistency are insufficiently justified (limited evidence/ablation and unclear evaluation rigor), the ML contribution is largely an application of conventional latent diffusion without clear methodological novelty, and the experimental setting relies heavily on small-scale synthetic data, making the generality of conclusions uncertain.

I align with the concerns and do not find the evidence strong enough for acceptance.

**Reviewer Concerns:**

Since the authors did not provide a response or clarification during the rebuttal, I believe most of the concerns raised by the reviewers remain unaddressed.

**Reviewer Scores:**

Since no rebuttal was provided, I expect that the reviewers’ scores will likely remain unchanged.

---

### Decision · Program_Chairs · 2026-01-26

Reject